# Dielectric Relaxation Behavior of Silver Nanoparticles and Graphene Oxide Embedded Poly(vinyl alcohol) Nanocomposite Film: An Effect of Ionic Liquid and Temperature

**DOI:** 10.3390/polym12020374

**Published:** 2020-02-07

**Authors:** Ganeswar Sahu, Mamata Das, Mithilesh Yadav, Bibhu Prasad Sahoo, Jasaswini Tripathy

**Affiliations:** 1School of Applied Sciences (Chemistry), Kalinga Institute of Industrial Technology, Bhubaneswar, Odisha 751024, India; titu.chem@gmail.com (G.S.); mamatadas1001@gmail.com (M.D.); 2Department of Chemistry, Prof. Rajendra Singh (Rajju Bhaiya) Institute of Physical Sciences for study and Research, V.B.S. Purvanchal University Jaunpur, Jaunpur 222003 U.P., India; dryadavin@gmail.com; 3Department of Chemical and Materials Engineering, Chang Gung University, Taoyuan 333, Taiwan

**Keywords:** dielectric permittivity, graphene oxide, poly(vinyl alcohol), polymer nanocomposite

## Abstract

This paper presents the dielectric characteristics of nanocomposite films of poly(vinyl alcohol) (PVA) embedded with silver (Ag) nanoparticles and graphene oxide(GO). The nanocomposite films were fabricated by using the solvent casting approach. The morphological analysis was carried out through scanning electron microscopy (SEM) and transmission electron microscopy (TEM). The dielectric relaxation behavior of nanocomposite films was analyzed in the frequency range of 10^1^ to 10^6^ Hz, by varying GO loading. The temperature effect was investigated over the temperature range of 40 to 150 °C. The effect of ionic liquid (IL) was also explored by comparing the dielectric behavior of films fabricated without using ionic liquid. The conductive filler loading variation showed a significant effect on dielectric permittivity(ε′), complex impedance (Z*) and electric conductivity (σ_ac_)_._ The obtained results revealed that the dielectric permittivity (ε′) increased by incorporating Ag nanoparticles and increasing GO loading in PVA matrix. An incremental trend in dielectric permittivity was observed on increasing the temperature, which is attributed to tunneling and hopping mechanism. With an increase in nanofiller loading, the real part of impedance (Z′) and imaginary part of impedance (Z″) were found to decrease. Further, the semicircular nature of Nyquist plot indicated the decrease in bulk resistivity on increasing GO loading, temperature and incorporating ionic liquid. On the basis of above findings, the obtained GO-Ag-PVA nanocomposite films can find promising applications in charge storage devices.

## 1. Introduction

Nanocomposites play an important role in the technological advancement of the 21st century. Nanocomposite is a multiphase material made up of matrix phase and the reinforcing phase, having enhanced thermal, mechanical, electrical, optical properties, as compared to individual phases. Due to the versatile nature of nanocomposites, they find numerous potential applications in various fields. [1,2,3,4]. Polymer nanocomposites are the combination of polymers as the matrix phase and the nanofillers like carbon-based nanoparticles, metal nanoparticles, etc. [5,6,7]. Polymer nanocomposites have attracted great interest of scientific communities due to their encouraging properties and widened applications, such as conductive coatings, sensors, energy storage devices, microwave absorbers and other devices [8,9,10,11,12,13]. Mostly, polymer composites with excellent dielectric characteristics are highly desirable for designing supercapacitors and electronic devices [14].

Recently, graphene and graphene derivatives have been extensively used for making the novel polymer nanocomposites, which have advanced one step ahead of the use of other conventional carbon nanofillers like carbon black and carbon nanotubes, due to their ultra-high aspect ratio, low density with excellent mechanical and dielectric properties, and thus find applications in energy storage devices [15,16]. Graphene, an allotrope of carbon is a two-dimensional monolayer sheet of sp^2^ hybridized carbon atoms densely packed in a honeycomb crystal structure that has revolutionized both the academic and industrial world to a greater extent. Compared to the carbon nanotubes, graphene is relatively cheaper due to its easy synthesis process. It exhibits a high level of electrical conductivity, extremely high carrier mobility and highly thermal conductivity with superior mechanical properties [17]. Graphene-based polymer nanocomposites exhibit outstanding mechanical and dielectric properties compared to other nanofillers [18,19,20]. Graphene oxide is the oxygenated graphene containing carboxyl, hydroxyl and epoxy groups at its edges, making it easier to disperse in water. The better interfacial bonding between GO and polymer matrix allows it to be compatible with many polymers for the formation of polymer nanocomposites [20]. Such polymer nanocomposites exhibit enhanced dielectric properties [21].

Poly vinyl alcohol (PVA) gained widespread attention, as it is water soluble, nontoxic, noncarcinogenic, biodegradable, biocompatible and optically transparent with high charge storage ability, and it also has wonderful film-forming attributes [22,23]. Due to the abundant hydroxyl groups, it is capable of resisting agglomerations with nanofillers and thus facilitates uniform dispersion of nanofillers in its matrix, and this is the key factor for the improved properties of the nanocomposites. PVA, being soluble in water, can be easily fabricated in an eco-friendly manner. The PVA-based nanocomposites can be exploited for electric applications by selecting suitable nanofillers. From the literature, it is also evident that PVA has a pivotal role in fuel cells, coating materials, opto-electronic devices, adhesives, fuel cells, transparent electrode materials, etc. [24,25,26,27,28,29]. These applications stimulate an interest in improving the mechanical, thermal and dielectric properties of PVA. The properties of the PVA can be modified by the incorporation of various nanofillers into it. The properties can be manipulated by various parameters, including extent of dispersion and polymer–filler interaction [6]. One of the methods of improving the dispersion of carbon based nanofillers in a polymeric matrix is by using suitable ionic liquids (IL). Ionic liquids are the salts with low melting point. They are the “green” alternatives of many organic solvents which could enhance many properties of polymer nanocomposites [7].

In the present scenario, materials with improved dielectric properties are in great demand. Silver nanoparticles have gained significant attention due to their interesting physical and chemical properties [30]. They find tremendous potential applications in nonlinear optics, electronics, catalysis, etc. [31,32,33,34]. It has been reported that Ag nanoparticles enhanced the dielectric properties of polymer nanocomposites [35]. In the present investigation, graphene oxide and Ag-embedded PVA nanocomposite films were fabricated by using the solvent casting technique and used to explore the dielectric behavior at different GO loading. The effect of ionic liquid and temperature have been explored on ε′, Z′, Z″, σ_ac_ and the Nyquist plot of the nanocomposite films.

## 2. Materials and Methods

### 2.1. Chemicals and Reagents

Poly(vinyl alcohol), AgNO_3_ and 1-allyl-3-methyl imidazolium chloride (AMIC) as ionic liquid were procured from Sigma-Aldrich (St. Louis, MO, USA). Potassium permanganate, sodium nitrate (NaNO_3_), sulphuric acid and hydrogen peroxide were purchased from S.D. Fine chemicals, India. All the chemicals and reagents procured were of analytical grade and used without further purification.

### 2.2. Synthesis of Graphene Oxide (GO)

Synthesis of GO was carried out by using a modified Hummer’s method [36]. In the synthesis process, 2 g of graphite powder and 1 g of sodium nitrate were dissolved in 46 mL of concentrated H_2_SO_4_ and stirred continuously for 30 min. To this mixture, 6 g of KMnO_4_ was gradually added, under a controlled temperature below 20 °C, followed by magnetic stirring for 2 h. Further, 96 mL of deionized water was poured into the suspension, keeping the temperature below 98 °C. After 15 min, 250 mL of warm deionized water was added to the reaction mixture. Thereafter, the removal of residual KMnO_4_ and MnO_2_ was carried out by mixing 20 mL of 30% H_2_O_2_. The sample was obtained by filtering and washing with 5% HCl and deionized water, followed by drying.

### 2.3. Nanocomposite Fabrication

#### 2.3.1. Fabrication of PVA and Ag-PVA Film

Fabrication of PVA, Ag-PVA and Ag-GO-PVA films was carried by using the solvent casting method. To prepare PVA film, 2.5 g of PVA was dissolved in 50 mL of deionized water and stirred continuously for 60 min, at 80 °C. After complete dissolution, the suspension was casted onto a Petri dish for drying, at room temperature. The film was obtained by peeling it from the petri dish. The silver-nanoparticles-embedded PVA nanocomposite was obtained by in situ synthesis. Briefly, AgNO_3_ of 5 mM concentration was added dropwise into PVA solution, with continuous stirring at 80 °C. The turning of the colorless solution into pale yellow indicated the synthesis of silver nanoparticles. Furthermore, the solution was casted onto a Petri dish, to obtain the film.

#### 2.3.2. Fabrication of GO-Ag-PVA and GO-Ag-PVA-IL Composite Film

The typical procedure for fabrication of GO-Ag-PVA is shown in Figure 1. In the fabrication process, GO suspension containing a prerequisite amount of GO in 25 mL water was sonicated for 1 h. The sonicated GO suspension was mixed with Ag-PVA solution and stirred magnetically for one hour. To obtain GO-Ag-PVA (IL) solution, ionic liquid 1-allyl-3-methyl imidazolium chloride (AMIC) was used. Then, 1 mmol of the ionic liquid was added into 1 g of GO. The GO-Ag-PVA (IL) nanocomposite was prepared by mixing Ag-PVA solution and GO suspension containing ionic liquid, under continuous magnetic stirring for one hour. The nanocomposite solution of three different weight percent were prepared by varying GO content with 1, 2 and 3 wt %. The solutions were casted on a Petri dish and dried at room temperature, to obtain the films. The nanocomposite films without IL were designated as 1GO-Ag-PVA, 2GO-Ag-PVA and 3GO-Ag-PVA, whereas the solutions containing ionic liquid were named as 1GO-Ag-PVA (IL), 2GO-Ag-PVA (IL) and 3GO-Ag-PVA (IL).

### 2.4. Characterization

The morphological analysis of films was carried out by using scanning electron microscopy (FEI-SIRION, Eindhoven, The Netherlands). Prior to the analysis, the samples were surface-coated with gold sputtering. The microstructural analysis of the composite films was performed, using transmission electron microscopy. Sample preparation was done by putting a drop of sonicated samples on Copper Grid. TEM images were captured by using JEM 1400 (JEOL Ltd., Tokyo, Japan).

### 2.5. Dielectric Relaxation Spectroscopy

The dielectric characteristics such as ε′, σ_ac_, Z′ and Z″ of PVA and its nanocomposite were analyzed, using a phase-sensitive multimeter (PSM1735N4L), over the frequency range of (1–10^6^ Hz), at room temperature, as well as by varying the temperature from 40 to 150 °C in an external alternating electric field.

The *σ_ac_* was obtained by using the following Equation:(1)$$\sigma_{ac}=\omega \varepsilon^{\prime }\varepsilon_{0}\tan\delta$$

Here, *ω* stands for the angular frequency of the applied external electric field.

The dielectric permittivity (*ε′)* and the absolute permittivity (*ε_ο_*) are obtained by using the following relation:(2)ε′=CpCo
where the *C_p_* corresponds to capacitance of the dielectric material, and *C_ο_* is the capacitance of vacuum.

## 3. Results

### 3.1. Morphological Analysis

The TEM micrographs of 2GO-Ag-PVA and 2GO-Ag-PVA(IL) nanocomposites are shown in Figure 2. The micrographs clearly show Ag nanoparticles as black dots embedded in graphene oxide sheets. The difference in dispersion pattern is clearly visible by comparing the TEM image of nanocomposite fabricated in absence (Figure 2a) and presence of ionic liquid (Figure 2b). The aggregated part is encircled in the TEM image (Figure 2c). Further, the effect of ionic liquid on dispersion is clearly distinct from the SEM micrographs. Figure 3a corresponds to 2GO-Ag-PVA nanocomposite film without ionic liquid which shows non-uniform dispersion of nanofillers with some aggregation, whereas the SEM micrograph (Figure 3b) of 2GO-Ag-PVA (IL) shows uniform morphology. These findings reveal that the use of ionic liquid increased dispersion of nanofillers in the PVA matrix. 

### 3.2. Dielectric Permittivity

The dielectric permittivity (ε′) signifies the ability of a material to store electric charge, and this depends upon charge, interfacial and dipole polarization under the influence of an external electric field. Polymer nanocomposites with high dielectric permittivity are used for energy storage and electronics applications [37,38,39]. Figure 4a exhibits the variation of ε′ of virgin PVA, Ag-PVA and GO-Ag-PVA nanocomposite films with the frequency of applied magnetic field at room temperature. It has been observed that ε′ is frequency dependent; however, the low frequency region has a greater effect on dielectric behavior. The virgin PVA exhibits significant ε′, and this could be due to its polar nature. Further, when Ag nanoparticles are embedded in PVA matrix, the dielectric permittivity is increased due to accumulation of charge at the interface of two materials which is commonly known as the Maxwell–Wagner–Sillars (MWS) effect [40]. The ε′ of GO-Ag-PVA is noticeably enhanced as compared to Ag-PVA, with a significant effect on decreasing frequency. The good compatibility of GO with PVA enhanced the dielectric properties of the PVA-Ag-GO nanocomposite. It is noteworthy that the dielectric permittivity increased with GO loading. The large surface area of GO facilitates the interfacial polarization developed at the junction of phases [41]. At lower frequency, the dipoles have enough time for interfacial polarization. However, at higher frequency, dipoles do not get sufficient time to orient themselves along the direction of applied fields. The rate of separation of charges is very low as compared to the rate of increase in frequency, thus exhibiting weak influence on dielectric behavior. The 3GO-Ag-PVA film exhibits its highest dielectric constant (ε′ = 30,378) at room temperature. The nanocomposite film with 2 wt % GO loading exhibits a maximum dielectric constant of 8796, whereas the nanocomposite film with 1 wt % GO loading shows a maximum dielectric constant of 3265.

Figure 4b represents the influence of GO loading, as well as IL on ε′ value of nanocomposite films over the entire frequency range. It is also compared with nanocomposite films fabricated without using IL. It is clearly visible that nanocomposite films fabricated using IL exhibit higher ε′ values with more pronounced effect at lower frequency. The enhanced ε′ value is due to the uniform distribution of nanofillers in the PVA matrix, resulting in easy interfacial polarization. Apparently, the ionic liquids are highly polar, thus nanocomposite fabricated with IL, possesses additional charge carriers, which facilitate interface polarization. IL reduces the nanosized gaps between the conductive channels, thereby increasing the value of ε′. Further, with an increase in the concentration of GO, dipole density increases, and this increases interfacial polarizability. At a lower frequency, the dipoles orient along the direction of the applied field; however, at a higher frequency, dielectric relaxation occurs due to the tremendous reduction in the diffusion of dipoles along the direction of the electric field [42]. The 3GO-Ag-PVA(IL) shows an ε′ value of 205,900, which is six times that of 3GO-Ag-PVA.

The effect of temperature (40 to 150 °C) on ε′ is shown in Figure 4c. It was observed that, with an increasing temperature, the value of ε′ increases. The increased temperature results in segmental motion of polarizable dipole [43]. In addition, increased temperature also initiates polymeric segmental motion, which narrows the gaps of conducting channels and thereby facilitates the polarization of dipoles [44]. The 1GO-Ag-PVA exhibits an ε′ value of 35,929 at 40 °C, and this value increases to 64,820,455 at 140 °C.

The effect of temperature is more pronounced at a lower frequency; however, at a higher frequency, dipoles fail to orient themselves along the direction of the electric field. The value of ε′ is 1804-fold at 150 °C as compared to 40 °C.

### 3.3. Impedance Analysis

To understand the complete conduction and relaxation mechanism of the heterogenous system, it is important to analyze the impedance characteristics. In an AC circuit, impedance corresponds to the resistive part of the material. It is the complex parameter given by the relation Z* = Z′ + iZ″, where Z′ represents the real part of impedance, and Z″ represents the imaginary part of impedance [45], which corresponds to the loss factor (reactance) of the system. The variation of Z′ with GO loading and frequency for virgin PVA, Ag-PVA and GO-Ag-PVA is shown in Figure 5a. The magnitude of Z′ gradually decreases with an increase in frequency, as well as filler loading, and this decrease can be explained on the basis of the frequency-dependent relaxation behavior of polymer nanocomposites. The relaxation phenomenon is dependent on physicochemical interactions between the nanofiller and polymeric phase. The conduction of current in the conductive nanocomposites is realized through continuous conductive channels. The homogeneous distribution of aggregated nanofillers facilitates the conduction of current. The Ag-PVA nanocomposite exhibits lesser Z′ value as compared to virgin PVA. Further, the introduction of carbon based conductive nanofiller GO offers less resistance to flow of current. The GO creates interlinked conductive pathways, which incline with increasing GO loading. Moreover, on increasing GO concentration, agglomerations become more prominent, decrease the nanoscopic gaps and thus facilitate the tunneling and hopping process, which is more significant in microwave frequency [46]. Interestingly, the systems exhibit low Z′ values at a lower frequency region mostly between 1 and 10^3^ Hz. At a lower frequency, dipoles get sufficient time to orient themselves along the field direction leading to the easy way of relaxation of dipoles, whereas, at high frequency, it is very difficult to orient dipoles. Moreover, low Z′ indicates absence of space charge polarization at higher frequency [45].

Figure 5b shows variation of Z′ with frequency for GO-Ag-PVA nanocomposites films fabricated in presence and absence of ionic liquid, and, further, it was compared with PVA and Ag-PVA films. A noticeable difference on dielectric relaxation behavior can be visibly seen due to the presence of ionic liquid, which can be explained on the basis of uniform distribution of nanofillers in the PVA matrix. The effect of temperature on Z′ is shown in Figure 5c. It can be clearly seen that the Z′ decreases continuously with an increase in temperature for 3GO-Ag-PVA. With an increment in temperature, segmental mobility of polymeric chain increases, and this influences the conductive nature of the system. At a higher frequency, the irregular decreasing behavior is observed, and it is mainly due to the loss factor of the system. The variation of Z″ with frequency at different filler loading, as well as in the presence of ionic liquid, has is shown in Figure 6a,b. There was an asymptotically decreasing behavior of Z″ with increasing frequency; in particular, a very low value of Z″ is seen at 3% GO loading in 3GO-Ag-PVA with drastic reduction in Z″ value in the case of 3GO-Ag-PVA(IL). Here, also, the same argument holds good as supported for variation of Z′ with frequency. A characteristic plateau can be observed for virgin PVA, as well as nanocomposite films, due to dielectric relaxation of the system. The peak position of plateau refers to the relaxation time which shifts toward right, due to an increase in relaxation time with conductive filler loading [47]. The peak gives valuable information about relaxation time of the dipoles. On increasing temperature, the peak position shifts toward the right (Figure 6c), which indicates the dielectric loss factor at a higher temperature [10].

### 3.4. Nyquist Plot

A Nyquist plot is obtained by plotting Z′ and Z″ on cartesian axes. It gives valuable information about the stability of dielectric material. It is very similar to Cole–Cole plot, which verifies dielectric characteristics of material. Figure 7a represents the Nyquist plots of virgin PVA, Ag-PVA and GO-Ag-PVA nanocomposite films. It is noteworthy that all the systems exhibit definite semicircular arcs between the frequency range of 1 to 10^6^ Hz.

These asymmetric semicircular arcs are indicative of non-Debye type of relaxation with different relaxation times [48]. The semicircular nature indicates the capacitive behavior of the system. It is observed that the diameter of the semicircular arc is shortened on increasing the filler loading. The center and radius of the semicircle signify the nanosized gaps between the conductive channels which are responsible for the non-ohmic conduction of electrons. The variation of filler loading has a remarkable effect on the real part and imaginary part of impedance. The real part of impedance corresponds to bulk resistance (R_B_), which is obtained from the intercept of semicircular plots on extrapolation along the *X*-axis. There has been a significant reduction of R_B_ values on increasing the conductive filler GO, and this signifies an increase in interlinked conductive channels.

The effect of ionic liquid on Nyquist plot is shown in Figure 7b. We noticed that the diameter of arc lessened significantly by incorporating ionic liquid. This behavior indicates uniform distribution of conductive nanofillers in the PVA matrix [49].

Figure 7c shows the effect of temperature on the Nyquist plot for 3GO-Ag-PVA. It is noteworthy that, on increasing temperature, the diameter of semicircular arc decreases, indicating the formation of some additional conductive channels. The intercept point which indicates R_B_ decreases with increasing temperature, indicating an improved conductive nature of 3GO-Ag-PVA. The diminished area of semicircular plot above 80 °C is presented in the inset of Figure 7c, which indicates the increased capacitive nature of the nanocomposite film at a higher temperature.

### 3.5. AC Conductivity

The variation of σ_ac_ with frequency for virgin PVA, Ag-PVA and GO-Ag-PVA nanocomposites have been shown in Figure 8a. PVA exhibits the lowest σ_ac_, whereas the conductivity of PVA-Ag is increased, as compared to the base polymer. Further, incorporation of GO shows an enhancement in σ_ac_ value. Interestingly, irrespective of GO loading, the electrical conductivity is increased with increase in applied frequency, having more prominent region between 10^4^ and 10^6^ Hz. At higher GO loading, a significant increase in σ_ac_ value is observed as compared to the lower GO loading. At lower loading, the conductivity can be explained by tunneling and hopping mechanisms [50]. The tunneling mechanism refers to the flow of charged particles through the continuous conducting channels on application of electric field, whereas, in hopping phenomena, the charged particles hop or jump between different conducting channels separated by nanosized gaps, as shown in Figure 9. The frequency-dependent incremental behavior of σ_ac_ is primarily attributed to hopping transport mechanism of charged particles. There is a remarkable increasing trend of σ_ac_ with increase in GO wt % in GO-Ag-PVA nanocomposite. The value of σ_ac_ is 8.95-, 17.5- and 35-fold for 1GO-Ag-PVA, 2GO-Ag-PVA and 3GO-Ag-PVA, respectively, in comparison to PVA. The GO and Ag nanofillers form interconnected conductive networks inside the PVA matrix that facilitate the hopping, as well as tunneling mechanism.

According to power law, in polymer nanocomposites, the relation between the angular frequency (ω) and σ_ac_ is given as follows:
*σ* (*ω*) ∞ *ω^s^*(3)
where “*s*” stands for the exponent, and the value of s lies between 0 and 1; however, the value of *s* is considered 1 in most of the cases.

The overall conductivity according to the AC universality law is as follows:(4)$$\sigma_{ac}\;=\;\sigma_{dc}\;+\;{\mathrm{As}}^{\omega }$$
where *σ_ac_* is equal to *σ_dc_*, when ω → 0, which explains the frequency-independent behavior of *σ_ac_*, where A and s are dependent on temperature [51].

The increasing trend of *σ_dc_* indicates the formation of interlinked conductive networks through which the transport of charge takes place. The σ_dc_ value increases with the increase in filler loading, as shown in Table 1. With an increment in filler loading, entangled network increases due to filler aggregates and thus forms the continuous network for the movement of charged particles. There has been a sharp rise in *σ_ac_* above 2 wt % GO loading. At higher conducting filler loading, a broad hopping rate is obtained due to heterogenous dispersion of fillers resulting into enhanced *σ_ac_* [44].

The effect of ionic liquid on *σ_ac_* of PVA-Ag-GO composites with different wt % of GO has been explored and compared with composite films without IL, as well as PVA-Ag and PVA matrix. The results are shown in Figure 8b. It is noteworthy that IL has a significant effect on the *σ_ac_* value of PVA-Ag-GO nanocomposite films. There has been an increase in value of σ_ac_ from 1.81 × 10^−6^ to 2.47 × 10^−6^ for 1GO-Ag-PVA (IL), 3.4 × 10^−6^ to 4.5 × 10^−6^ in case of 2GO-Ag-PVA (IL) and 6.9 × 10^−6^ to 3.62 × 10^−5^ for 3GO-Ag-PVA (IL) in comparison to nanocomposite films without IL. The *σ_ac_* value is dependent on degree of dispersion of the nanofiller. The ionic liquid serves as a coupling agent between GO and PVA that facilitates the hopping and tunneling phenomenon of charged particles [52].

Figure 8c shows the variation of *σ_ac_* of 3GO-Ag-PVA with frequency at different temperature. It is distinctly visible that *σ_ac_* values increase with an increase in temperature, which corresponds to negative temperature co-efficient (NTC) of resistance [53]. The increased *σ_ac_* with increasing temperature can be due to thermal activation of nanofillers, leading to the formation of conductive networks. Moreover, heating of polymer nanocomposites leads to aerial oxidation of polymer in presence of carbon nanofiller, resulting in the formation of polar carbonyl groups. The increased conductivity can be explained by percolation theory, electron tunneling theory and electric field radiation theory.

The percolation theory considers the transport of charged particles from one end to another end of conductive network on an application of applied electric field [54]. According to electron tunneling theory, conduction takes place by tunneling or hopping [55]. As per the radiation theory, the emission current generated by applying applied field flows between conducting materials, which are separated by nanoscopic gaps [43]. Apart from that, the increased *σ_ac_* with increasing temperature can also be explained on the basis of segmental motion of polymeric units [54].

## 4. Discussion

In summary, multifunctional hybrid polymer nanocomposites based on PVA-Ag-GO were successfully fabricated by using solvent casting technique. The dispersion pattern of the Ag and GO nanoparticle has been evaluated from SEM and TEM photomicrographs. The effect of Ag, GO and the IL on the dielectric relaxation behavior of the fabricated PVA based nanocomposites was well explored over a wide range of frequency of applied field and temperature. The increase in highly polarizable dipoles by embedding Ag nanoparticles and GO in PVA matrix resulted in significant improvement in εʹ from 6 × 10^2^ to 3 × 10^4^. The establishment of interconnected conductive networks in the hybrid nanocomposites has been confirmed from the substantial increase in *σ_ac_* up to a value of 3.62 × 10^−5^ for 3GO-Ag-PVA (IL) due to high extent of tunneling and hopping mode of conduction mechanism. The modification in the interfacial zone of the nanocomposites at higher temperature was confirmed from the noticeable improvement in all the dielectric properties. The easiness in the polarization of the dipoles at high temperature is responsible for the increase in ε′ and substantial increase in *σ_ac_*, whereas noticeable decrease in Z′ and Z″ at higher temperature confirms the negative temperature coefficient (NTC) behavior of the developed nanocomposites. The modified PVA-based nanocomposite may find promising practical applications in energy-storing devices.

## Figures and Tables

**Figure 1 polymers-12-00374-f001:**
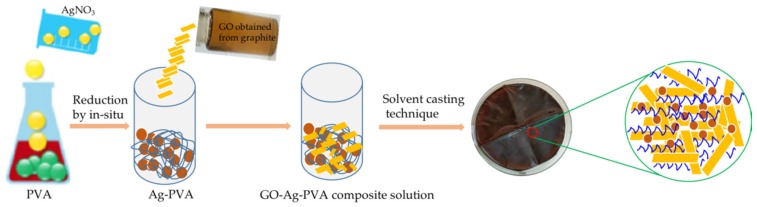
Schematic presentation of fabrication of GO-Ag-PVA nanocomposite film.

**Figure 2 polymers-12-00374-f002:**
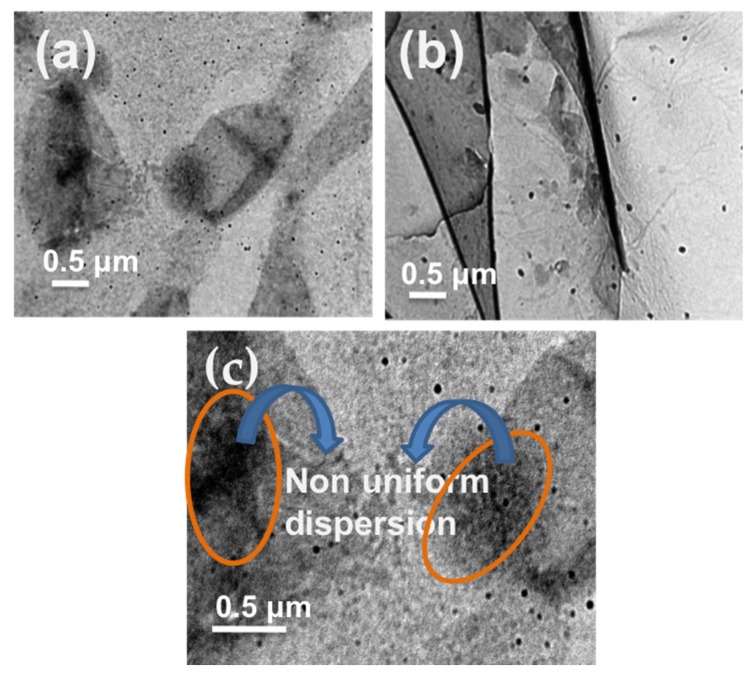
TEM images of (**a**) GO-Ag-PVA, (**b**) GO-Ag-PVA (IL) and (**c**) 2GO-Ag-PVA.

**Figure 3 polymers-12-00374-f003:**
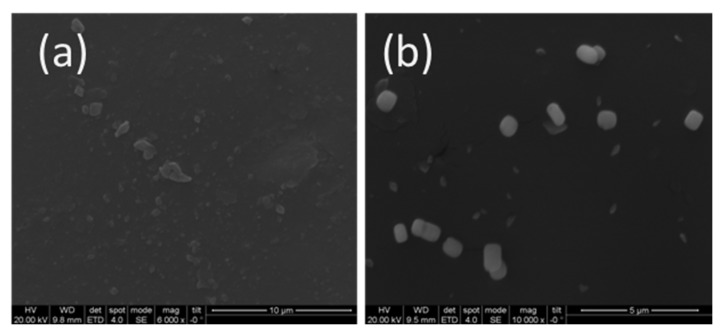
SEM images of (**a**) 2GO-Ag-PVA (**b**) 2GO-Ag-PVA (IL).

**Figure 4 polymers-12-00374-f004:**
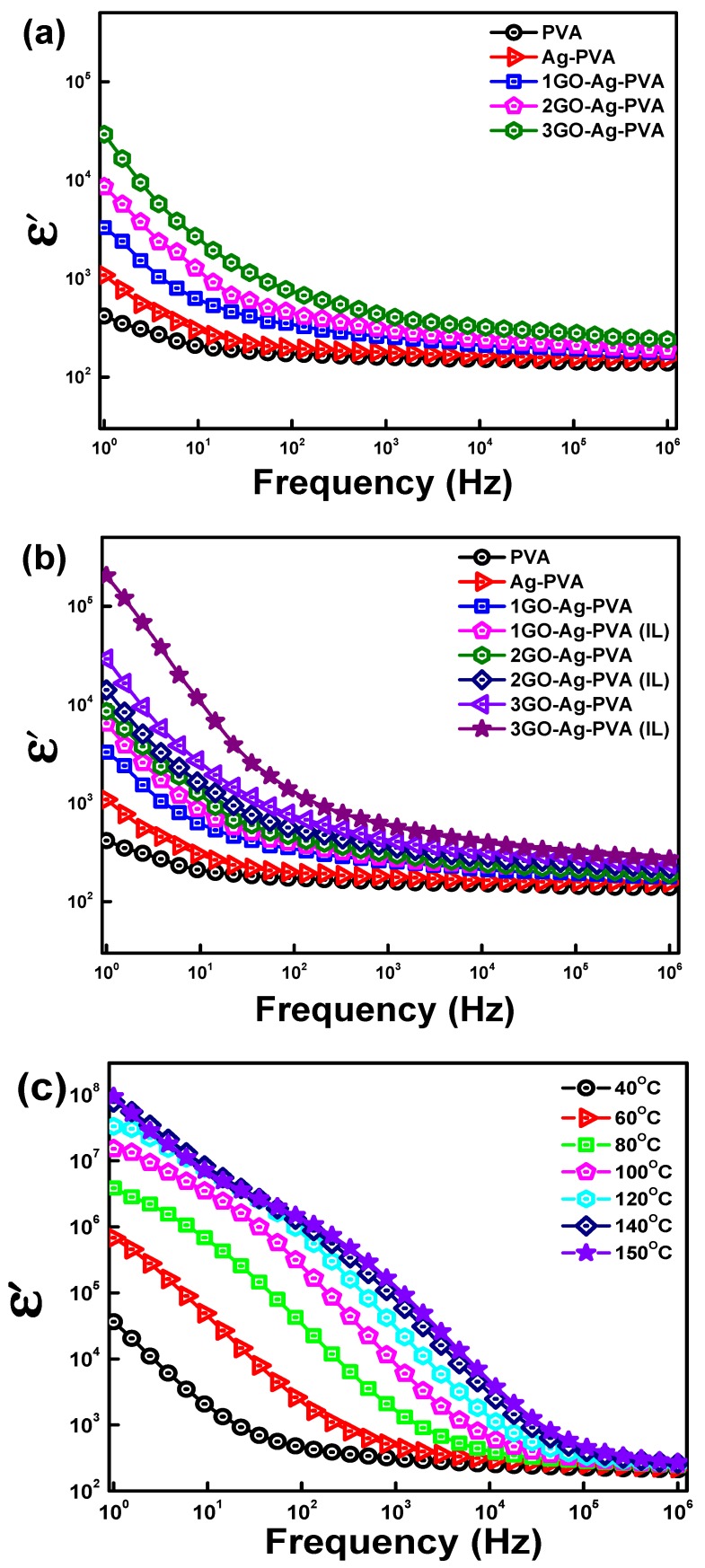
Dielectric permittivity of GO-Ag-PVA nanocomposites as a function of (**a**) GO loading, (**b**) ionic liquid and (**c**) temperature effect on 3GO-Ag-PVA (IL).

**Figure 5 polymers-12-00374-f005:**
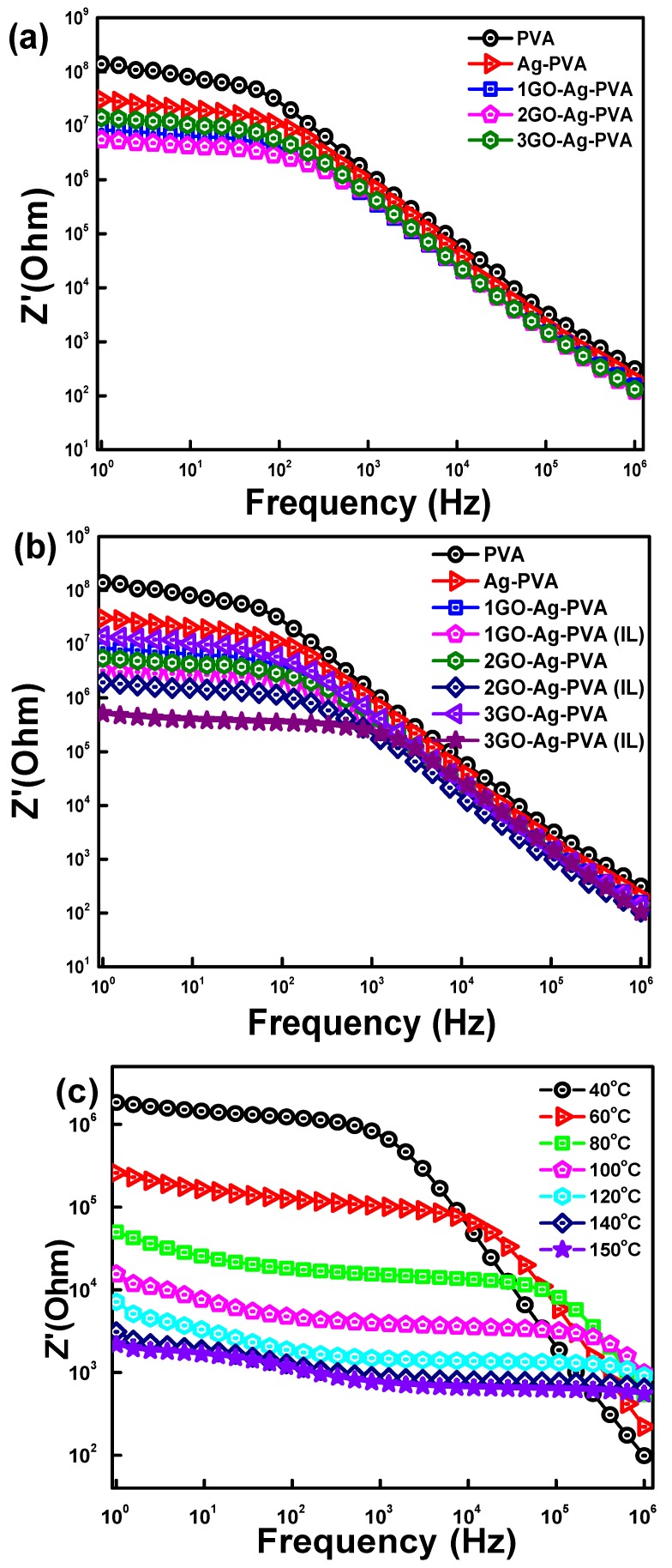
Real impedance of GO-Ag-PVA nanocomposites as a function of (**a**) GO loading, (**b**) ionic liquid and (**c**) temperature effect on 3GO-Ag-PVA (IL).

**Figure 6 polymers-12-00374-f006:**
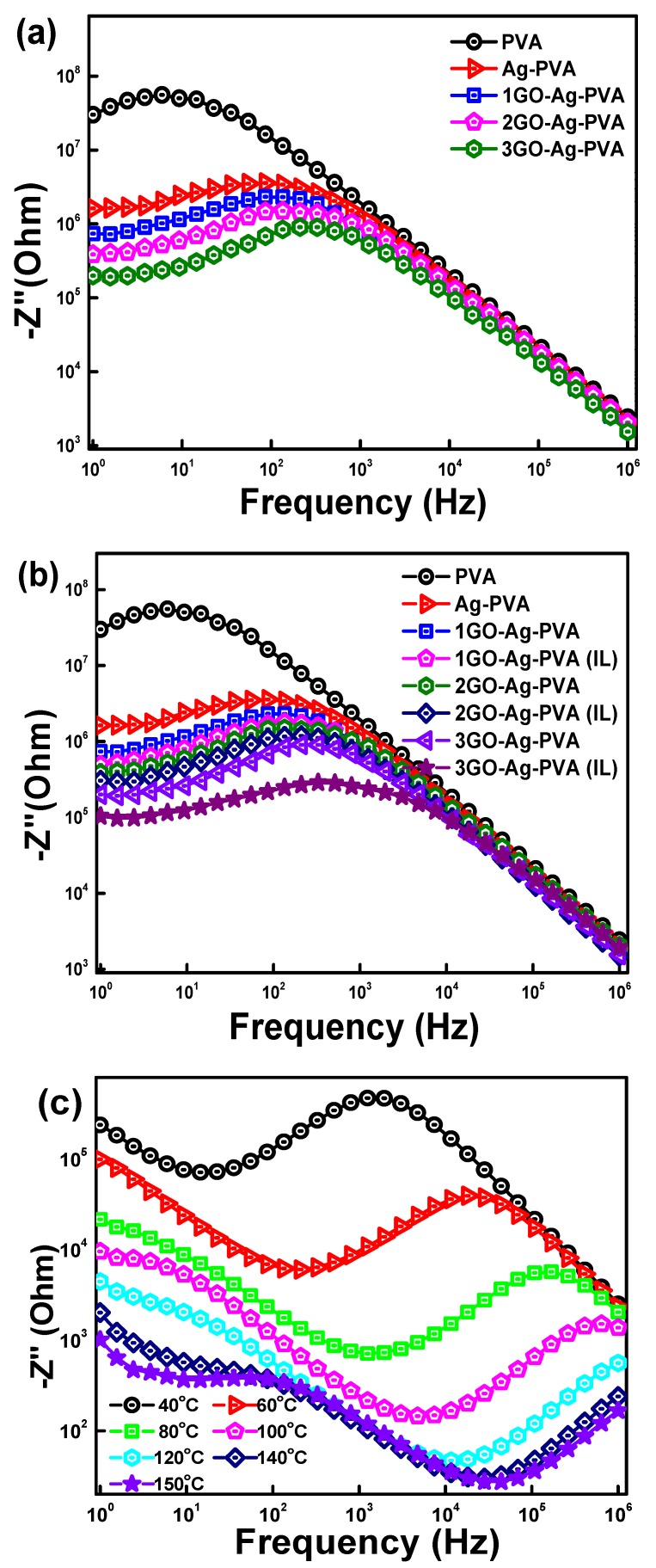
Imaginary impedance of GO-Ag-PVA nanocomposites as a function of (**a**) GO loading, (**b**) ionic liquid and (**c**) temperature effect on 3GO-Ag-PVA (IL).

**Figure 7 polymers-12-00374-f007:**
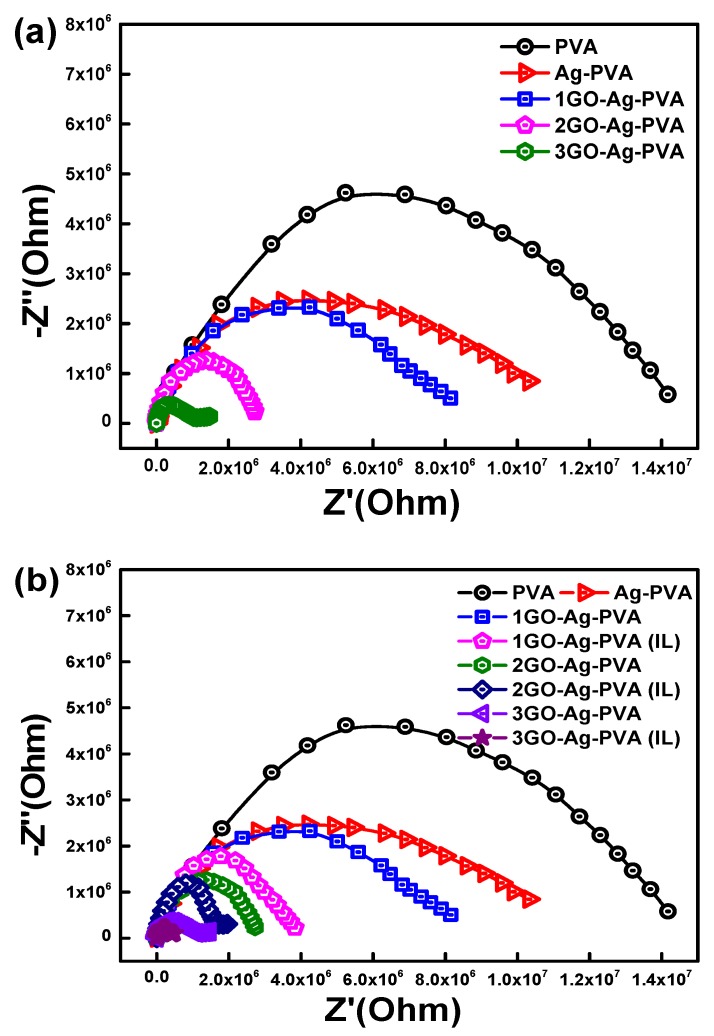
Nyquist plot of GO-Ag-PVA nanocomposites as a function of (**a**) GO loading, (**b**) ionic liquid and (**c**) temperature effect on 3GO-Ag-PVA (IL). The inset shows magnified image of (c)

**Figure 8 polymers-12-00374-f008:**
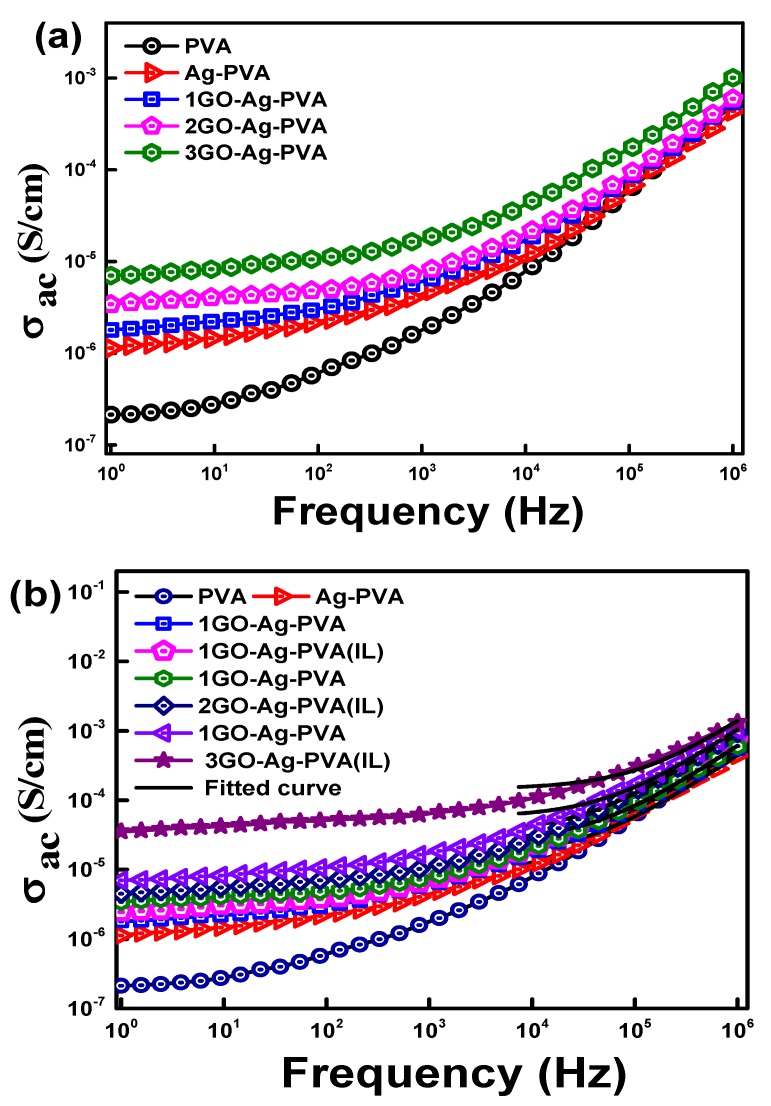
AC conductivity of GO-Ag-PVA nanocomposites as a function of (**a**) GO loading, (**b**) ionic liquid and (**c**) temperature effect on 3GO-Ag-PVA (IL).

**Figure 9 polymers-12-00374-f009:**
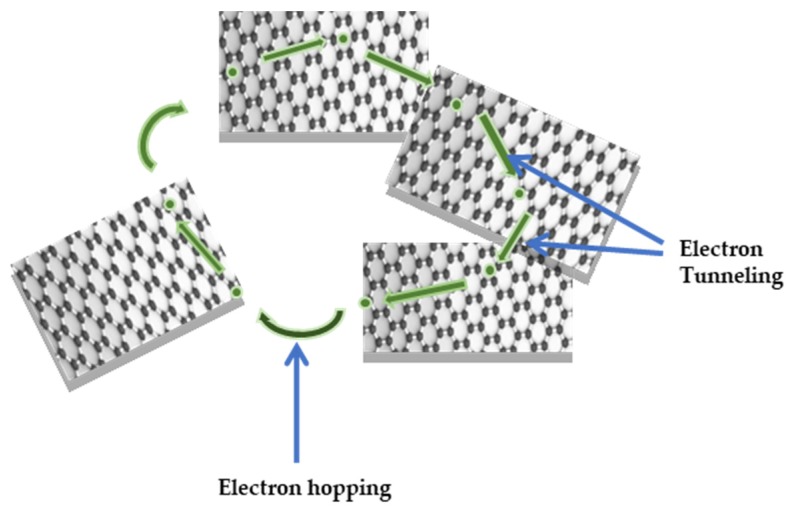
Schematic representation of tunneling and hopping mechanism of electrons.

**Table 1 polymers-12-00374-t001:** DC conductivity values obtained from the power law fitting curves of the conductivity plots.

Sample Codes	σ_dc_ (S/cm)	A	s
PVA	7.65 × 10^−6^	1.70 × 10^−6^	4.73 × 10^−5^
Ag-PVA	1.54 × 10^−5^	1.37 × 10^−4^	4.98 × 10^−7^
1GO-Ag-PVA	2.17 × 10^−5^	1.44 × 10^−4^	5.89 × 10^−7^
1GO-Ag-PVA(IL)	2.22 × 10^−5^	8.37 × 10^−5^	1.48 × 10^−6^
2GO-Ag-PVA	2.52 × 10^−5^	2.81 × 10^−6^	3.31 × 10^−5^
2GO-Ag-PVA(IL)	3.41 × 10^−5^	4.17 × 10^−5^	3.49 × 10^−6^
3GO-Ag-PVA	5.68 × 10^−5^	1.17 × 10^−6^	1.34 × 10^−4^
3GO-Ag-PVA(IL)	1.46 × 10^−4^	1.28 × 10^−4^	1.55 × 10^−6^

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
