# Peer review of "Dielectric Relaxation Behavior of Silver Nanoparticles and Graphene Oxide Embedded Poly(vinyl alcohol) Nanocomposite Film: An Effect of Ionic Liquid and Temperature"

_polymers, 2020, doi:10.3390/polym12020374_

Round 1
Reviewer 1 Report
To my opinion, the authors have presented the materials in a coherent and reasonable manner. This article should be published in Polymers Journal.
However, few comments and concerns are mentioned below.
1. In line 43 and 44, the authors commented: "Mostly, polymer composites with excellent dielectric characteristics are highly desirable for designing supercapacitors and electronic devices.". I think this statement should be referenced.
2.In line 245,the word 'relation' should read 'relaxation'. Line 246, the peak shift towards"..."as shown in fIG. 6c.
2. It was shown that added ionic liquid helps in getting a better dispersion of the fillers within PVA matrix,which is supported by SEM image and their corresponding impedance measurements spectra. The statement regarding that made in a qualitative manner. Is there any way,quantitative measures can be deduced from charge transport? e.g. the mean distance between the two conductive channels?
3. What about the molecular weight effect in the overall nanocomposite performance?
4. Why ionic liquids help with the dispersion of nanofiller?Is there any chemistry is responsible between GO and ionic liquids? will a hydrophobic polymer in this regard behave in the same way?
Author Response
Response to Reviewer 1
To my opinion, the authors have presented the materials in a coherent and reasonable manner. This article should be published in Polymers Journal.
However, few comments and concerns are mentioned below.
In line 43 and 44, the authors commented: "Mostly, polymer composites with excellent dielectric characteristics are highly desirable for designing supercapacitors and electronic devices.". I think this statement should be referenced.
Response: As suggested by learned reviewer, reference for the statement "Mostly, polymer composites with excellent dielectric characteristics are highly desirable for designing supercapacitors and electronic devices." is introduced as ref no- 14.
In line 245,the word 'relation' should read 'relaxation'. Line 246, the peak shift towards"..."as shown in fig. 6c.
Response: The word 'relation' has been corrected as "relaxation" and the line corresponding to peak shift has been completed and rewritten.
It was shown that added ionic liquid helps in getting a better dispersion of the fillers within PVA matrix, which is supported by SEM image and their corresponding impedance measurements spectra. The statement regarding that made in a qualitative manner. Is there any way, quantitative measures can be deduced from charge transport? e.g. the mean distance between the two conductive channels?
Response: The ionic liquid helps in exfoliation of graphene oxide sheets as reported in literature "Liu et al. Nanoscale, 2018, 10, 8115–8124". Author has been carried SEM and TEM to show uniform dispersion. Further, the same is supported by impedance measurements experiments. We have not carried any experiments to find out the mean distance between conductive channels however it can be carried during TEM analysis of samples.
What about the molecular weight effect in the overall nanocomposite performance?
Response: Author agreed the molecular weight of polymer is important factor for film fabrication purpose. In case, if we use different PVA (different molecular weight), the mechanical performance of the film will vary. The dielectric performance of the nanocomposites depend on polarizable dipoles. So, any factor which enhances the polarizable dipoles will have an impact on dielectric properties. Author will try to focus on this type of work in next project. Here, author humbly request to reviewer to accept this work as such. In this project author used PVA with low molecular weight. In all the fabricated films, the molecular weight and weight of polymer is same. However, nanofiller loading percentage is varied.
Why ionic liquids help with the dispersion of nanofiller? Is there any chemistry is responsible between GO and ionic liquids? Will a hydrophobic polymer in this regard behave in the same way?
Response: Ionic liquids help in dispersion and preparation of nanomaterials as reported in literature. The explanation is described below:
The amine group of ionic liquid forms bond with carboxylic acid group of GO through amide bond. The interaction of GO and ionic liquid is well studied before and reported in various research papers. It helps in exfoliation of GO sheets thus GO get uniformly distributed in polymers. The ionic liquid also serves as a coupling agent between GO and PVA which facilitates the hopping and tunnelling phenomenon of charged particles as given in reference [53].The hydrophobic polymers can also be used. Exfoliated GO can be dispersed in hydrophobic polymers through pi-pi interaction.

Reviewer 2 Report
Reviewers comments:
Sahu et al., fabricated GO-Ag-PVA composite films with IL and without IL successfully. The dielectric relaxation behaviour with/without IL and temperature effect was well studied. The presented SEM and TEM of materials are well incorporated for justifying its characterization. Further, the semi-circular nature of Nyquist plot showed the decrease in bulk resistivity on increasing GO loading, temperature and incorporating ionic liquid. Overall, this work is interesting and novel. But, still so many queries should be resolved before its publication. Therefore, minor revision is needed for manuscript.
Authors can introduce few more references related to your work (AgNP)- Materials Science and Engineering: C, 53 (2016) 36-43; (Generalized polymer composite) Composites Part B 73 (2015) 166-180; (Dielectric property) Composites Part B, 160 (2019) 632-643; (Graphene composites)Journal of Nanomaterials, 2013 (2013), Article ID 763953, 14 pages. Line 80-81, Nanocomposites…….researchers needs corrections. Please rearrange the sentence of line 90-91 and 99-100 in proper way. Gram and hour should be replaced by “g” and “h” in appropriate places within the manuscript. Line 118-120, 121-123 and 134-135 needs rechecking. Please modify it. In figure 1 and 2, the captions are not provided in following journal guideline. Check and modify it. Fig2 authors kindly improve the quality of the scale bar. Authors should provide an XRD with library indexed peaks and EDXA either for SEM /TEM. Please provide Raman spectrums so as to determine the GO -PVA-Ag interactions and stability. The provided equations in manuscript should be numbered and all their symbols and variables should be defined. Author should take care of it. In some figures; e.g. 4a, 4b,5a,5b,6a,6b,8a and 8b; the provided “FREQUENCY” word should be replaced by “frequency”. Please check and modify it. In Line 231-232, author missed some figure numbering. So, explanation of results is not justified. Please check Nyquist plot fig 7c inset, authors should explain the inset plot more clearly. In Line 309-310, what do author wants to infer with the equation? Please recheck and modify it. In Table, author should use multiplication symbol “×” instead of “X”. The line 359-362 of discussion part should be rearranged. The “Discussion” part should be more technical.
Author Response
Response to Reviewer 2
Sahu et al., fabricated GO-Ag-PVA composite films with IL and without IL successfully. The dielectric relaxation behaviour with/without IL and temperature effect was well studied. The presented SEM and TEM of materials are well incorporated for justifying its characterization. Further, the semi-circular nature of Nyquist plot showed the decrease in bulk resistivity on increasing GO loading, temperature and incorporating ionic liquid. Overall, this work is interesting and novel. But, still so many queries should be resolved before its publication. Therefore, minor revision is needed for manuscript.
Authors can introduce few more references related to your work (AgNP)- Materials Science and Engineering: C, 53 (2016) 36-43; (Generalized polymer composite) Composites Part B 73 (2015) 166-180; (Dielectric property) Composites Part B, 160 (2019) 632-643; (Graphene composites) Journal of Nanomaterials, 2013 (2013), Article ID 763953, 14 pages.
Response: The new references have been introduced in the manuscript as suggested by learned reviewer.
Line 80-81, Nanocomposites…….researchers needs corrections.
Response: Corrections have been implemented as "It has been reported that Ag nanoparticles enhanced the dielectric properties of polymer nanocomposites”.
Please rearrange the sentence of line 90-91 and 99-100 in proper way.
Response: The lines have been rewritten and arranged as "All the chemicals and reagents procured were of analytical grade and used without further purification".
Gram and hour should be replaced by “g” and “h” in appropriate places within the manuscript.
Response: Gram is replaced by g and hour by h as suggested by esteemed reviewer.
Line 118-120, 121-123 and 134-135 needs rechecking. Please modify it.
Response: Suggested lines have been modified in the manuscript.
In figure 1 and 2, the captions are not provided in following journal guideline. Check and modify it.
Response: In Figure 1 and 2, the captions have been arranged corrected according to journal guideline.
Figure 2 authors kindly improve the quality of the scale bar.
Response: Correction has been implemented accordingly.
Authors should provide an XRD with library indexed peaks and EDXA either for SEM /TEM. Please provide Raman spectrums so as to determine the GO-PVA-Ag interactions and stability.
Response: Thanks for valuable suggestion. Author did not provide Raman and XRD in the manuscript as the purpose of study is to focus on of dielectric behaviour of polymer nanocomposite. But, in next project both characterizations will be inserted. Therefore, author humbly request to reviewer to accept manuscript as such.
The provided equations in manuscript should be numbered and all their symbols and variables should be defined. Author should take care of it.
Response: Equations in manuscript have been numbered and all the symbols and variables have been defined properly.
In some figures; e.g. 4a, 4b,5a,5b,6a,6b,8a and 8b; the provided “FREQUENCY” word should be replaced by “frequency”. Please check and modify it.
Response: Figures have been modified and uniformity is maintained accordingly.
In Line 231-232, author missed some figure numbering. So, explanation of results is not justified.
Response: Author has inserted the numbering of figure for proper explanation of results accordingly.
Please check Nyquist plot fig 7c inset, authors should explain the inset plot more clearly.
Response: As suggested by esteemed reviewer, the explanation for inset is provided in text more clearly.
In Line 309-310, what do author wants to infer with the equation? Please recheck and modify it.
Response: Corrections have been implemented in equation.
Table, author should use multiplication symbol “×” instead of “X”. The line 359-362 of discussion part should be rearranged. The “Discussion” part should be more technical.
Response: Modified table and rearranged discussion part with more technical way were provided in the text accordingly.
